# A View-consistent Sampling Method for Regularized Training of Neural Radiance Fields

## Abstract

Neural Radiance Fields (NeRF) has emerged as a compelling framework for scene representation and 3D recovery. To improve its performance on real-world data, depth regularizations have proven to be the most effective ones. However, depth estimation models not only require expensive 3D supervision in training, but also suffer from generalization issues. As a result, the depth estimations can be erroneous in practice, especially for outdoor unbounded scenes. In this paper, we propose to employ view-consistent distributions instead of fixed depth value estimations to regularize NeRF training. Specifically, the distribution is computed by utilizing both low-level color features and high-level distilled features from foundation models at the projected 2D pixel-locations from per-ray sampled 3D points. By sampling from the view-consistency distributions, an implicit regularization is imposed on the training of NeRF. We also propose a novel depth-pushing loss that works in conjunction with the sampling technique to jointly provide effective regularizations for eliminating the failure modes. Extensive experiments conducted on various scenes from public datasets demonstrate that our proposed method can generate significantly better novel view synthesis results than state-of-the-art NeRF variants as well as different depth regularization methods.

## 1 Introduction

3D scene reconstruction from multiple images is a long-standing vision problem (Hartley and Zisserman, 2000) but the recent advent of Neural Radiance Fields (NeRFs) (Mildenhall et al., 2020) has delivered a significant performance boost, especially given a dense set of input images. However, in much the same way the old shape-from-shading was ill-posed (Prados and Faugeras, 2005), so is the NeRF reconstruction problem: as shown in (Zhang et al., 2020), in the absence of explicit or implicit regularization, a set of training images can be fitted independently of the recovered geometry. This phenomenon, known as shape-radiance ambiguity, is particularly evident when the input views are not dense enough, even though using Multi-Layer Perceptrons (MLPs) for scene representation weakly regularizes the scene reconstructions (Zhang et al., 2020; Yu et al., 2022).

Many kinds of regularizers have been proposed to improve on this, such as imposing geometric constraints (Kim et al., 2022; Niemeyer et al., 2022), training to directly predict radiance fields by using networks conditioned on image features (Chen et al., 2021; Yu et al., 2021; Wang et al., 2021), or constraining depth (Deng et al., 2022; Wang et al., 2023a; Yu et al., 2022). The first is difficult to do for complicated scenes while the second is often limited to very specific 3-view setting and not easily generalizable to unbounded scenes or scalable to more input views. The last, depth regularization, has proved to be the more widely applicable. However, it typically requires expensive 3D supervision, and can produce unreliable predictions on challenging open-space scenes that produce artifacts in the final NeRF reconstruction.

To remedy this, we propose to use view-consistency distributions per-ray instead of fixed depth predictions to implement a sampling technique along the rays that implicitly regularizes the training of NeRF, as depicted by Fig. 1. Specifically, we first distill geometric information from the redundant feature representations of foundation models to reduce their dimensionality and to alleviate memory requirements, while preserving the information most likely to be consistent across views and discarding the rest. Given these high-level distilled features along with low-level color features, we compute a view-consistency metric for every point along the rays and introduce an adaptive sampling

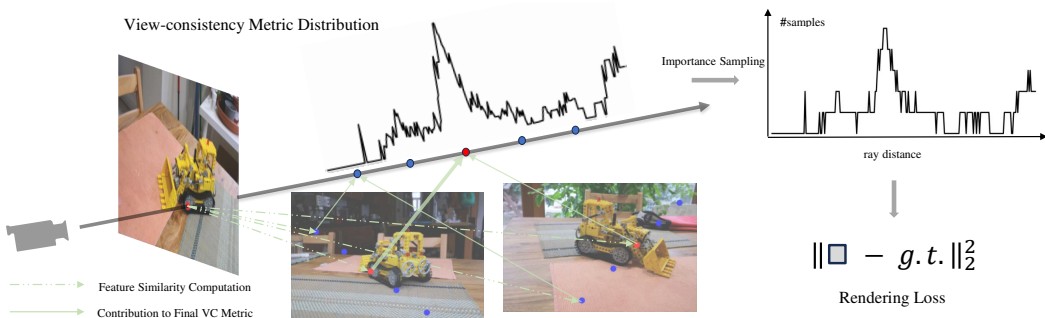

Figure 1: **View-consistent sampling.** Our central idea is to pre-compute a view-consistency distribution along rays and to perform importance sampling according to this distribution. As a result, the sampling will concentrate around surface points instead of random points in the capture volume.

scheme that favors view-consistent points, on the assumption that they are more likely to lie on a real-world surface. Furthermore, we also introduce a depth-pushing loss to force the model to favor samples that are farther away from the camera origin, which prevents the kind of background collapse artifact (Barron et al., 2022) that frequently happens in NeRF reconstruction of real-world unbounded scenes. In effect, the proposed view-consistent sampling and the depth-pushing loss focus the NeRF reconstruction process on the part of the capture volume close to the true surface, thus providing implicit regularization and preventing the overfitting problem (Zhang et al., 2020).

Our contribution is therefore a novel view-consistent sampling technique to implicitly regularize the training of NeRF, along with a depth-pushing loss to provide further regularization and mitigate background collapse artifacts. We show that our method is able to achieve significantly better novel view synthesis results compared to existing NeRF competitors with regularizations. Our implementation is based on open-source software and will be made publicly available.

## 2 RELATED WORKS

**NeRF Variants.** The emergence of Neural Radiance Fields (NeRF) (Mildenhall et al., 2020) is an immediate consequence of the study on Implicit Neural Representations (INR) (Tancik et al., 2020; Sitzmann et al., 2020; Hertz et al., 2021; Mehta et al., 2021), which laid the foundation of NeRF by introducing powerful network-based scene representation models. NeRFs use them to effectively encode 3D scene properties and trains on posed images via a volume rendering equation that differentiably relates 3D scenes to 2D images.

NeRFs deliver outstanding image synthesis results but tendsto be slow, sometimes requiring several days for a single scene. A number of accelerated approaches have therefore been proposed (Sun et al., 2022; Fridovich-Keil et al., 2022; Chen et al., 2022; Müller et al., 2022), using various kinds of efficient scene representation techniques. Interestingly, not only are these approaches faster, they also tend to yield better image synthesis results. Other works have focused more directly on improving the quality of the synthesis results (Zhang et al., 2020; Barron et al., 2021; 2022; 2023; Turki et al., 2024), by introducing unbounded scene representations or reducing aliasing artifacts. Nerfacto (Tancik et al., 2023), introduced in the popular Nerfstudio project, combines many components of these approaches into an integrated one. Hence, this is what we use as the basis for implementing our own approach.

**Regularizers.** A straightforward approach to improving the performance of NeRFs is to incorporate geometric priors to regularize and guide the training process. To this end, many methods have been proposed. They rely on depth guidance (Deng et al., 2022; Wang et al., 2023a;b; Roessle et al., 2022; Yu et al., 2022), geometric constraints (Niemeyer et al., 2022; Somraj et al., 2023; Truong et al., 2023; Kim et al., 2022), or pre-training on similar scenes (Chen et al., 2021; Yu et al., 2021; Wang et al., 2021; Wu et al., 2023; Xu et al., 2023). However, these methods are all plagued by generalization issues. For depth priors, it is difficult to obtain accurate depth predictions, especially in real-world unbounded scenes. The geometry-based constraints often fail to properly refine the results in complex unbounded scenes. Prediction-based methods are mostly restricted to a 3-view setting in bounded scenes, due to the limitations of a prediction-based architecture and the limited

availability of real-world unbounded scene data with 3D ground truth. Recently, ReconFusion (Wu et al., 2024) has been proposed to incorporate diffusion piror into NeRF training. This method works well for indoor or bounded scenes, but for open-space scenes the performance drops drastically as the original paper shows.

**Image Features.**    Recently, there has been tremendous progress in large-scale self-supervised pre-training using Masked image Modeling (He et al., 2022; Wei et al., 2022; Zhou et al., 2021) (MIM). These new techniques provide us with foundation models, such as DINOv2 (Oquab et al., 2023), which encode local geometric information better than classification pretraining (Xie et al., 2023) and generalize well to geometric vision tasks, e.g. image geometric matching (Sun et al., 2021; Edstedt et al., 2023). While there are also models specifically designed and trained for image geometric matching, their pairwise matching setting is ill-suited to NeRFs dealing with image collections because using them would involve traversing all image pairs.

## 3 METHODOLOGY

We now introduce our *View-consistent Sampling* (VS-NeRF) approach for NeRF training. As our method is built on the standard NeRF framework, we first describe it briefly in Sec. 3.1. Next, we describe our approach to distill high-level image features and preserve only view-consistent information from the foundation model DINOv2 (Oquab et al., 2023) in Sec. 3.2. We then introduce a sampling mechanism to exploit these features as well as color features along camera rays in Sec. 3.3. Finally, we describe the proposed depth-pushing loss as a weaker regularization to force the model to favor distant samples in in Sec. 3.4.

### 3.1 NERF BASICS

**Scene Representation.**    The 3D scene is generally represented by an MLP, and optionally and additional feature grid, which encode both geometry information and view-dependent color information. Specifically, the geometry of the scene is encoded by the neural network as a function $f : \mathbb{R}^3 \to \mathbb{R}$ that maps a spatial coordinate $\mathbf{x} \in \mathbb{R}^3$ to its corresponding volume density value $\sigma$. The view-dependent color information is encoded by the network as a function $f : \mathbb{R}^3 \times \mathbb{S}^2 \to \mathbb{R}^3$ that takes a point coordinate $\mathbf{x} \in \mathbb{R}^3$ as well as a viewing direction $\mathbf{d}$ as input and outputs the associated view-dependent color value $\mathbf{c} = (r, g, b)$.

**Volume Rendering.**    The rendering process is of critical importance because it associates a 3D representation of the scene with 2D images, which makes the use of image reconstruction loss possible. In the NeRF literature, the most frequently used rendering technique in 3D vision tasks is known to be volume rendering. Given a ray $\mathbf{r}(t) = \mathbf{o} + t\mathbf{d}$, the volume rendering equation yields the color of one pixel in the 2D image corresponding to the ray $\mathbf{r}$ by evaluating

$$\hat{\mathbf{C}}(\mathbf{r}) = \int_{t_n}^{t_f} \omega(t)\mathbf{c}(\mathbf{r}(t), \mathbf{d})dt , \qquad (1)$$

where $\omega(t) = T(t)\sigma(\mathbf{r}(t))$ is the weight function, $\sigma$ represents the volume density, $\mathbf{c}$ represents the directional color, and $T(t) = \exp(-\int_{t_n}^{t} \sigma(\mathbf{r}(s))ds)$ represents the transparency function.

In practice, this integral is evaluated by sampling the ray in discrete locations. NeRF volume rendering is then performed by accumulating color values from $S$ samples $(t_i)_{1 \le i \le S}$ along a ray $\mathbf{r}$. This yields

$$\hat{\mathbf{C}}(\mathbf{r}) = \sum_{i=1}^{S} T_i(1 - \exp(-\sigma_i\delta_i))\mathbf{c}_i, \text{ where } T_i = \exp\left(-\sum_{j=1}^{i-1} \sigma_j\delta_j\right) , \qquad (2)$$

where $\delta_i = t_{i+1} - t_i$ is the distance between two consecutive samples.

**Loss Function.**    Given the estimated color $\hat{\mathbf{C}}(\mathbf{r})$ of Eq. 2, let $\mathbf{C}(\mathbf{r})$ be the corresponding pixel true color. Using these notations, we can define an MSE loss

$$\mathcal{L}_{color} = \frac{1}{|\mathcal{B}|} \sum_{\mathbf{r} \in \mathcal{B}} \|\hat{\mathbf{C}}(\mathbf{r}) - \mathbf{C}(\mathbf{r})\|^2 , \qquad (3)$$

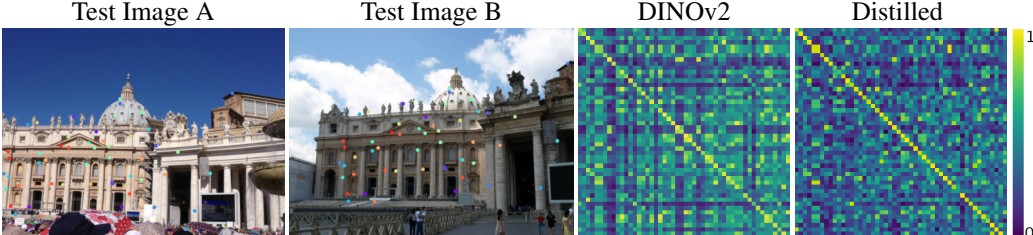

Figure 2: Visualization of the feature distillation process. For the two test images from Megadepth dataset (Li and Snavely, 2018), we first randomly generate 50 ground truth correspondences (same as in the training process), shown as colored dots, and then extract vanilla DINOv2 features (384 dimension) and the proposed distilled DINOv2 features (32 dimension) at these locations. We compute the feature similarities across the two views and show the resulting similarity matrices on the right, where an optimal correspondence should give the identity matrix.

where $\mathcal{B}$ denotes a randomly chosen batch of rays and $|\mathcal{B}|$ denotes the batch size. The weights of the NeRF scene representation network are computed by minimizing this loss, using a different batch of rays at each iteration.

### 3.2 Distillation of Geometric Information.

To form meaningful view-consistent statistics for adaptive sampling, a good representation of images in the context of multi-view captures is critical. In this paper, we propose to use foundation models that provide powerful general-purpose visual features, e.g. DINOv2 (Oquab et al., 2023), for extracting such high-level representations containing crucial context information. Note that there are alternatives to DINOv2 such as features from diffusion models (Luo et al., 2024), which we found to deliver similar results. Since diffusion models are slower in inference time, we opted to use DINOv2. However, there are two main hurdles in utilizing the features from foundation models. Firstly, as general-purpose features, the output of a foundation model encodes image information in many different aspects that are useful in different tasks. In the NeRF setting, we are particularly interested in geometric information that can be expected to be similar across multi-view images of the same scene. Secondly, the dimensionality of features from foundation models is in general very high, e.g. 384 in DINOv2, resulting in prohibitively large memory consumption in sampling.

In this paper, we propose to resolve the issues by distilling geometric information from the foundational features. Note that we use the term *distillation* in a different way than in *Feature Field Distillation* papers, such as (Kobayashi et al., 2022), which focuses on lifting 2D features to a 3D representation. We distill features by extracting geometric information from redundant high-dimensional image features. Inspired by (Luo et al., 2024), we use a very lightweight Resnet bottleneck block (He et al., 2016) to project the high-dimensional features to a lower dimension for distillation. To supervise the distillation process, we adopt the Megadepth dataset (Li and Snavely, 2018) which provides 3D ground truth and is prevalently used in geometric matching tasks.

**Training of Distillation Process.** We adopt a very simple strategy for training. Specifically, in the training phase, we freeze the foundation model and only update the parameters in the Resnet bottleneck block. This leads to a much smaller number of training parameters and also requires much less data. We randomly choose 50000 pairs of images from Megadepth to train. For each image pair, we use the ground truth depth map to randomly generate 50 corresponding points, and extract the distilled features at the point locations on the image pairs. We then supervise the network using a symmetric cross entropy loss, in the same fashion as CLIP (Radford et al., 2021), to make extracted features in corresponding locations as close as possible while still being distinctive from other features. A visualization of the distillation process can be found in Fig. 2. As a result, in our experiments we can reduce the feature dimensionality by a factor of around 10, e.g. 384 from DINOv2 to 32, without compromising on useful geometric information. Please refer to the ablation studies in Sec. 4.2 for the discussion of optimal dimensionality in NeRF settings.

| Reference Point | VC Distribution | Reprojection A | Reprojection B | Reprojection C |

Figure 3: Visualization of the effectiveness of the view-consistency metric on the BONSAI scene from the MipNeRF360 dataset. As shown, we shoot a ray from the reference point in the leftmost image, compute the view-consistency metric distribution along the ray, and reproject the peak point in the distribution onto other views. The projections of the peak are consistent and correspond to a surface point.

## 3.3 VIEW-CONSISTENT SAMPLING

In the original NeRF and most NeRF variants, the volume rendering of Eq. 2 is typically achieved using naive sampling strategies such as uniform, stratified, or linear disparity sampling. Hence, there is no prior in the sampling process and hence no regularization while learning the radiance and density fields. When there are abundant input views, this is usually not an issue but it can result in unwanted artifacts with a smaller set of input images.

VS-NeRF remedies this by making the sampling adaptive based on a prior: it samples more densely the 3D locations whose projections have view-consistent features because they are more likely to correspond to 3D surface points. This requires both a view-consistency metric and an adaptive sampling scheme based on that metric, which we describe in Sections 3.3.1 and 3.3.2, respectively. A graphical visualization of the proposed view-consistent sampling technique can be seen in Fig. 1.

**Sampling Setup.** We assume that we have a collection of $N$ posed images $\{\mathbf{I}_i\}_{i=1}^N$, from which we generate image feature representations $\{\mathbf{F}_i\}_{i=1}^N$. As in NeRF and most of its variants, the training is performed by repeatedly sampling a batch of rays. For each ray $\mathbf{r}$, we initially need to place pre-samples along the ray $(t_i)_{1 \leq i \leq M}^{pre}$, to obain $M$ points $\{\mathbf{p}_i\}_{i=1}^M$. Note that these pre-samples are different from the initial samples in NeRF, as pre-samples are for computing view-consistency statistics only and will not be used to compute losses. The pre-samples are generated uniformly within a distance, but after a fixed threshold the step sizes will increase with each sample due to scene contraction. This strategy is the same as the initial sampling strategy in Nerfacto (Tancik et al., 2023) and we refer to the original paper for details.

### 3.3.1 COMPUTATION OF VIEW-CONSISTENCY METRIC

**Features from Projections.** As shown in Fig. 3, the feature representation at the pixel location where the ray comes from is denoted as reference feature $\mathbf{f}_r$. For each pre-sample point $\mathbf{p}_i$, we can project it onto an arbitrary view $v_j$. If the point $\mathbf{p}_i$ is visible to $v_j$, then naturally by interpolating over the feature representation $\mathbf{F}_j$, we can obtain the projection feature $\mathbf{f}_{ij}$. Due to limited field-of-view (FOV) of cameras, there is a varying number of views that a point $\mathbf{p}_i$ can be projected onto. We denote the set of views that a point $\mathbf{p}_i$ can be projected onto as $V_i$, and $|V_i|$ as its cardinality.

**Normalized Similarity Measure.** In this paper, we jointly use high-level distilled features, and plain normalized RGB values as low-level color features. Please see the ablation study in Sec. 4.2 to understand their respective effects. However, the two kinds of features are defined in different metric spaces. That is to say, while Euclidean distances can be used for measuring discrepancies among color features, cosine similarities are most frequently used as a distance measure of features from pre-trained models such as DINOv2. In this paper, we use a normalizing strategy to convert the measures among features to binary numbers, be it color feature or distilled feature. In particular, along an arbitrary ray $\mathbf{r}$, we first compute the measures between the reference feature and projection features from all sampled points $\{m(\mathbf{f}_r, \mathbf{f}_{ij}) \mid j \in V_i\}$, be it Euclidean distances or cosine similarities. We then normalize the set of measures based on its mean and variance, and take the negative if the measure is Euclidean distance to align distance with similarity. Thus, we have defined the normalized similarity measure $\{m_n(\mathbf{f}_r, \mathbf{f}_{ij}) \mid j \in V_i\}$.

**View-consistency Metric.** Given the the normalized similarity measure $\{m_n(\mathbf{f}_r, \mathbf{f}_{ij}) \mid j \in V_i \}$, we assume its values follow a normal distribution and we experimentally determine a reasonable threshold $\delta$ accordingly. The view-consistency metric of point $\mathbf{p}_i$ along the ray is computed as:

$$s_i = \frac{1}{|V_i|} \sum_{j \in V_i} \mathbb{1}\{m_n(\mathbf{f}_r^c, \mathbf{f}_{ij}^c) > \delta \ \wedge m_n(\mathbf{f}_r^d, \mathbf{f}_{ij}^d) > \delta\} \ , \tag{4}$$

where $\mathbb{1}$ denotes the indicator function, superscripts $c$ and $d$ denote *color* and *distilled* in projection features respectively. Intuitively, Eq. 4 measures the average view consistency over the views the point can be projected onto. Although occlusions may hinder the effectiveness of this metric, statistically the score is still prominent for surface points. A visualization of the computed view-consistency metrics along a ray using real data can be seen in Fig. 3.

### 3.3.2 ADAPTIVE SAMPLING SCHEME

Given the view-consistency metric of Eq. 4, implementing view-consistent sampling becomes straightforward. After computing the metric for each pre-sampled point along the ray, we perform importance sampling based on the distribution along the ray. Our rationale is that this view-consistentcy distribution is concentrated around the surface point, thus importance sampling from the distribution is the logical way to improve it.

In our implementation, we use the Probability Distribution Function (PDF) sampler from Nerfstudio (Tancik et al., 2023) to perform importance sampling, which generatse samples that match a distribution. Specifically, as illustrated by Fig. 1, we first compute view-consistency metrics from pre-samples $(t_i)_{1 \le i \le M}^{pre}$ along the ray. The PDF sampler will probabilistically sample the bins between consecutive pre-sample points, such that the distribution of number of samples in each bin will match the view-consistency distribution, which gives the true samples $(t_i)_{1 \le i \le S}$ to compute losses as in Eq. 2.

### 3.4 DEPTH-PUSHING LOSS

In NeRF, the background of the scene is generally harder to reconstruct than foreground objects, typically because parts of the background may only be seen in very few views. This can result in *background collapse*, a notorious NeRF artifact that erroneously creates false geometries near the camera for background objects. The view-consistent sampling scheme of Section 3.3 mitigates this problem but it can still occur in challenging cases because the feature representations extracted from background pixels are often less reliable due to perspective effects. Thus, to complement our adaptive sampling scheme, we introduce a depth-pushing loss

$$\mathcal{L}_{depu} = -\frac{1}{|\mathcal{B}|} \sum_{\mathbf{r} \in \mathcal{B}} \log(d(\mathbf{r}) + \varepsilon), \text{ where } d(\mathbf{r}) = \sum_{i=1}^{S} T_i(1 - \exp(-\sigma_i \delta_i))t_i \ , \tag{5}$$

where $\varepsilon$ is a small constant that stabilizes the logarithm function near 0 and $d(\mathbf{r})$ is the expected depth along the ray. Minimizing $\mathcal{L}_{depu}$ favors distant samples along the ray and provides a regularization to prevent background collapse. Its simple form makes it easy to integrate into the NeRF framework by adding it to the color loss of Eq. 3.

## 4 EXPERIMENTAL RESULTS

In this section, we demonstrate the effectiveness of our VS-NeRF approach, which includes a discussion of experimental settings and implementation details; evaluation results on benchmark datasets and comparison with previous work; an ablation study with respect to the major components in VS-NeRF; and a discussion of limitations.

**Implementation Details.** Our implementation of VS-NeRF is built upon the Nerfacto method from the Nerfstudio project (Tancik et al., 2023). It incorporates many published methods that have been found to work well for real data, such as Mip-NeRF360 (Barron et al., 2022), IntantNGP (Müller et al., 2022), and NeRF-W (Martin-Brualla et al., 2021). We simply replace Nerfacto's sampling scheme by ours and add our depth-pushing loss and keep all other settings the same. Notably, the

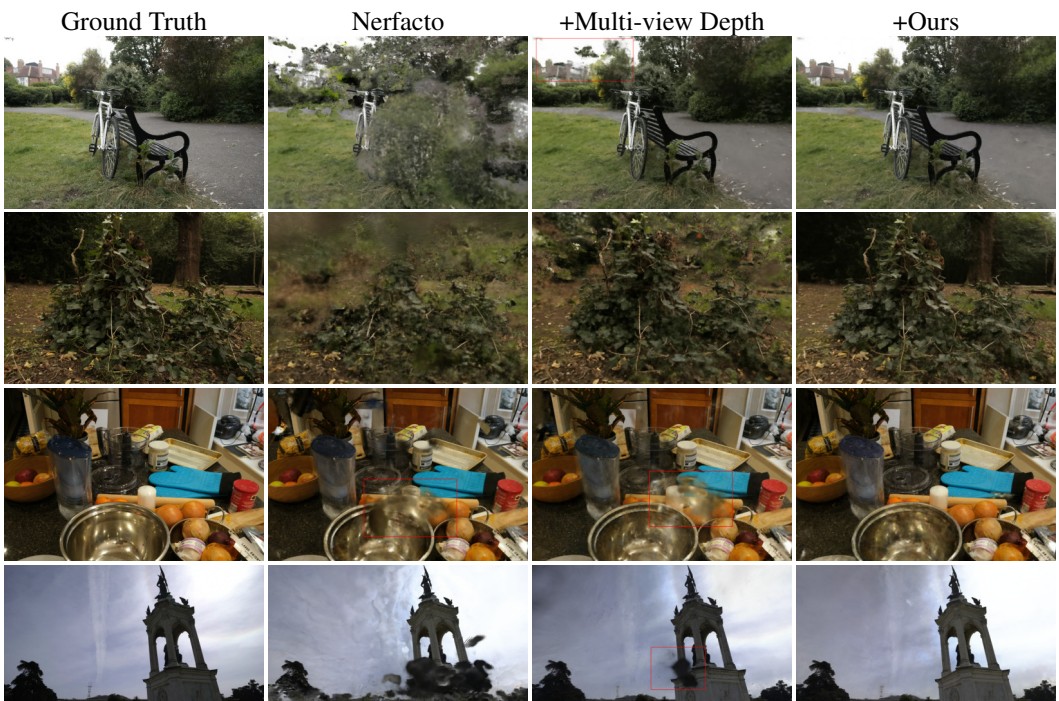

Figure 4: We show comparisons of VS-NeRF to the main competitors and the corresponding ground truth images from held-out test views. The scenes are, from the top down: BICYCLE with 60 training views, STUMP with 110 training views, COUNTER with 70 training views from the Mip-NeRF360 dataset and FRANCIS with 70 training views from Tanks&Temples. The '+' prefix indicates the included additional component to Nerfacto.

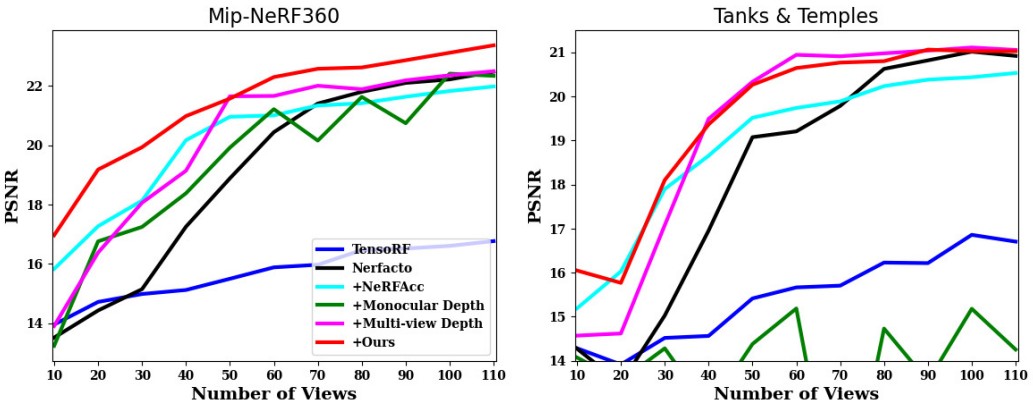

Figure 5: We show performances of VS-NeRF and competitors with increasing number of views over Mip-NeRF360 dataset and Tanks&Temples, in terms of PSNR values. The '+' prefix indicates the included additional component to Nerfacto.

Nerfacto proposal network sampling scheme, from Mip-NeRF360 (Barron et al., 2022), is also left unchanged. This ensures that any difference in performance is attributable to our regularization scheme. We turn off the camera optimization for both VS-NeRF and Nerfacto, since we observed a negative impact on datasets with accurate camera parameters.

To reduce the time cost, we only activate the proposed view-consistent sampling technique in the first 5000 iterations out of 30000 in total, which is when the regularization of the geometry is the most needed. Empirically, the threshold $\delta$ in Eq. 4 is set to $0.4$, the weight of depth-pushing loss is set to $0.0001$, and the $\varepsilon$ in the depth pushing loss as in Eq. 5 is set to $0.01$.

| Dataset | Mip-NeRF360 | | | | Tanks&Temples | | | |
|---|---|---|---|---|---|---|---|---|
| Method / Metric | PSNR↑ | SSIM↑ | LPIPS↓ | Train | PSNR↑ | SSIM↑ | LPIPS↓ | Train |
| TensoRF | 15.68 | 0.455 | 0.658 | 25m29s | 15.46 | 0.609 | 0.566 | 29m07s |
| Nerfacto | 19.05 | 0.549 | 0.495 | 11m37s | 18.29 | 0.688 | 0.422 | 11m16s |
| +NeRFAcc | 20.14 | 0.580 | 0.480 | 12m18s | 18.95 | 0.691 | 0.431 | 12m07s |
| +Monocular Depth | 19.45 | 0.549 | 0.488 | 11m55s | 13.81 | 0.451 | 0.632 | 12m20s |
| +Multi-view Depth | 20.15 | 0.578 | 0.465 | 12m24s | 19.28 | 0.706 | 0.397 | 12m10s |
| +Ours | 21.40 | 0.625 | 0.400 | 38m44s | 19.45 | 0.714 | 0.373 | 39m05s |

Table 1: Quantitative evaluation of our method compared to previous work, computed over two datasets. The '+' prefix indicates the included additional component to Nerfacto.

**Baselines.**    Since our implementation is based on **Nerfacto** (Tancik et al., 2023), we treat it as a baseline to demonstrate the positive impact of our adaptive sampling scheme and depth-pushing loss. We also use **TensoRF** (Chen et al., 2022) as another baseline that features efficient training. In addition we also compare against InstantNGP (Müller et al., 2022) from the Nerfstudio project. The most prominent difference between InstantNGP (Müller et al., 2022) and **Nerfacto** (Tancik et al., 2023) is the **NeRFAcc** (Li et al., 2023) efficient sampling scheme, whose name we will use to refer to this method.

Regarding depth regularizations, we compare against both monocular and multi-view methods, using the depth-nerfacto method, again from Nerfstudio project. To test the method with **Monocular Depth** regularization, we use ZoeDepth (Bhat et al., 2023) to predict pseudo depth and apply depth-ranking loss from SparseNeRF (Wang et al., 2023a). To test the method with **Multi-view Depth** regularization, we use the state-of-the-art MVSFormer++ (Cao et al., 2024) to provide depth estimations from correlating with adjacent 9 views, along with the depth loss from DS-NeRF (Deng et al., 2022).

**Datasets and Metrics.**    We use two benchmark datasets for our main evaluation, first the 9 full scenes from Mip-NeRF360 (Barron et al., 2022) and second all 8 scenes from the INTERMEDIATE official test set in Tanks & Temples dataset (Knapitsch et al., 2017). The scenes in the two datasets contain both a complex central object or area and a detailed background, and cover both bounded indoor scenes and large unbounded outdoor environments, making them challenging for NeRF methods. We use the same hyperparameter configuration for all experiments.

In order to study the effect of the number of available views on the reconstruction quality, we subsample between 10 to 110 images per scene, 110 being the size of the smallest image set in our datasets. To this end, for each scene, we first evenly sample 10 images as an evaluation set and then sample evenly the remaining views. In ablation study, we use 50 views for all scenes, as it is a reasonable number for practical usage and we observed that the need for regularization is highest as the number of views decreases.

Following the usual convention, we report quantitative results based on PSNR, SSIM (Wang et al., 2004), and LPIPS (Zhang et al., 2018), along with the training time in minutes as measured on a single NVIDIA A100 80GB GPU.

## 4.1    COMPARATIVE RESULTS

We report our comparative results on our two datasets as a function of the number of training views being used in Fig. 5. We provide evaluation metrics averaged over all different scenes with different training view numbers in Tab. 1 and qualitative results in Fig. 4.

VS-NeRF clearly outperforms all baselines in terms of novel view synthesis quality on Mip-NeRF360. On Tanks&Temples, multi-view depth regularization is on par with our method when views are dense, but our method performs better in sparser cases. Crucially, our method significantly outperforms Nerfacto, upon which our implementation is based, which conclusively demonstrates the effectiveness of our view-consistent sampling and depth-pushing loss. Note that NeRFAcc also yields considerable improvement over Nefacto when available views are sparser but its relative performance drops when the number of views increases, which is not the case for our approach. TensoRF seems to

| | PSNR ↑ | SSIM ↑ | LPIPS ↓ | Train |
|---|---|---|---|---|
| A) Base Method Nerfacto | 18.88 | 0.544 | 0.503 | 13m20s |
| B) Base + VS (Sec. 3.3) | 19.34 | 0.574 | 0.456 | 38m54s |
| C) Base + DL (Sec. 3.4) | 17.09 | 0.478 | 0.567 | 11m27s |
| D) Base + VS (Only Color Feature (Eq. 4)) + DL | 19.42 | 0.571 | 0.454 | 32m51s |
| E) Base + VS (Only Distilled Feature (Eq. 4)) + DL | 19.96 | 0.584 | 0.434 | 38m11s |
| F) Base + VS (Distilled Feature Dimension as 16 (Sec. 3.2)) + DL | 20.04 | 0.587 | 0.432 | 39m22s |
| G) Base + VS (Distilled Feature Dimension as 64 (Sec. 3.2)) + DL | 19.97 | 0.585 | 0.433 | 48m22s |
| Base + VS + DL (Our Complete Model) | 21.57 | 0.631 | 0.400 | 38m44s |

Table 2: An ablation study in which we remove or replace the major components in our method to measure their effect on the Mip-NeRF360 dataset with 50 training views. The two major components are VS: *View-consistent Sampling* and DL: *Depth-pushing Loss*.

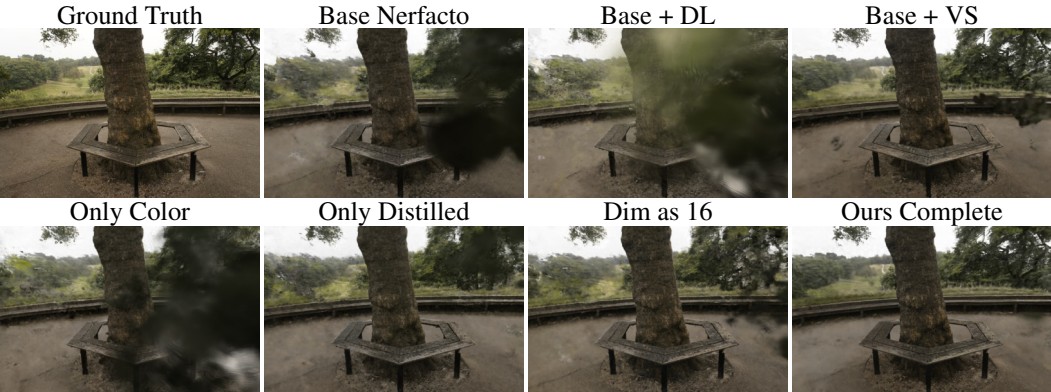

Figure 6: Visualization of ablating major components in our method, using the TREEHILL scene from Mip-NeRF360 dataset.

struggle most, presumably due to the limitations of tensor-based scene representation in complex and unbounded scenes.

We also compare agains two different depth-based regularizers, a monocular one using MVSformer++ and a multi-view one using ZoeDepth. Monocular depth estimation produces many artifacts and results in unstable performance, especially on Tanks&Temples. This is largely because monocular depth is only a pseudo depth that may not be consistent across views. The multi-view depth regularization performs significantly better than the monocular one and consistently improves over Nerfacto. However, in challenging scenes with few available views it may produce unreliable depth estimations. In contrast, our method samples from a view-consistency distribution and is more robust as can be seen in results on Mip-NeRF360 dataset.

## 4.2 ABLATION STUDY

We perform an ablation study of the two novel components of our approach, i.e. the *View-consistent Sampling* (VS) and the *Depth-pushing Loss* (DL). For simplicity, the experiment is conducted with a moderate 50-view setting for subsampling the scenes on Mip-NeRF360 dataset. The results are presented in Tab. 2 and a visualization is given in Fig. 6. The first three rows of the table show that each component, VS or DL, brings an improvement when used separately. Comparing to our complete model in ie last row, it shows that they work best when used jointly.

Regarding the features used to compute the view-consistency metric, distilled DINOv2 features are more powerful than color feature when used independently, but the best performance is achieved by combining them, as in Eq. 4. The dimensionality of the distilled features is also investigated here. We see that reducing the dimensionality to 16 or increasing it to 64 will degrade performance. Thus, we opt to use 32 as the dimensionality of distilled features in our implementation.

## 4.3 LIMITATIONS

Despite the excellent results in visual quality delivered by VS-NeRF, it has some limitations. This includes shrinking effectiveness with available views increasing as shown in Fig. 5, and efficiency issues. Specifically, we see that VS-NeRF takes more time than the efficient competitors from Tab. 1, and it also requires more memory. For example, for a scene with around 80 input images, it will consume roughly 25 GB memory. However, since our method is most effective when the views are less dense, this drawback of a concern in practice. Furthermore, more efficient implementations are possible and will be explored.

## 5 CONCLUSION AND DISCUSSIONS

In this paper, we have proposed a novel view-consistent sampling technique as a regularization for the training of NeRF. The core idea is to combine high-level and low-level features to compute view-consistency metrics, and use it as a prior distribution to sample on the ray. To mitigate the background collapse problem, we also propose a depth-pushing loss, which imposes a weaker regularization to favor distant samples on the ray. Extensive experiments on public datasets have demonstrated the effectiveness of the proposed method.

**Broader Impacts.** The method in this paper can help generate highly realistic 3D scenes from 2D images, which can find applications in various fields such as education and entertaining. On the other hand it may also be used to create realistic forgeries which requires careful considerations.

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

# A   APPENDIX / SUPPLEMENTAL MATERIAL

## A.1   APPROXIMATE STORAGE COMPUTATION

The storage problem of projection features is addressed by our proposed distillation process as in Sec. 3.2, here we show the necessity of it. For a batch of rays with size $|\mathcal{B}|$, the tensor to store the projection features would be of size $|\mathcal{B}| \times M \times N \times C$, where $M$ is the number of sampled points per ray, $N$ is number of input views, and $C$ is the feature dimension. This is prohibitive when the feature dimension $C$ is very large, for example, in the common setting where $|\mathcal{B}| = 4096$, $M = 256$, $N = 50$ and $C = 384$, it will consume roughly 80GB memory with float datatype. But if we distill the features such that $C = 32$, the memory requirement becomes roughly 6.7GB.

