# OpenReview forum: "A View-consistent Sampling Method for Regularized Training of Neural Radiance Fields"
_ICLR.cc/2025/Conference — ICLR 2025 Conference Withdrawn Submission_

### Official Review · Reviewer_pcyv · 2024-10-29

**Soundness:** 3
**Presentation:** 3
**Contribution:** 3
**Rating:** 5
**Confidence:** 4

**Summary:**

The paper proposes a regularization method to improve point sampling along rays when training NeRF. The method’s core idea is a view-consistent sampling technique, which distills geometric features from the DINOv2 foundation model to learn a view-consistent ray distribution, allowing for more efficient point sampling. Additionally, the paper introduces a depth-pushing loss to favor distant points, addressing issues with false geometry. Experimental results show that this approach enhances Nerfactor’s performance, surpassing several depth regularization methods.

**Strengths:**

- The quantitative results in Table 4 and the qualitative results in Figure 4 clearly demonstrate the proposed method’s improvements.
- The two novel components -- view-consistent sampling and depth-pushing loss -- are sound and well-motivated.
- Overall, the paper is well-written.
- The proposed regularizations are compatible with existing methods, benefiting the field of NeRF-based models.

**Weaknesses:**

- Since the paper proposes an efficient ray sampling technique, it should compare to other existing efficient sampling techniques, such as Coarse-to-Fine Online Distillation in Mip-NeRF360 (CVPR '22) and Probabilistic Ray Sampling in SceneRF (ICCV '23).

- The impact of the distilled features' quality on rendering performance is unclear. Since these features are learned to match points across images, the paper should analyze the quantitative performance of point matching and its influence on downstream results.

- The existing ablation study is poorly done. Particularly, it lacks of explanation for each row in Tab 2. Furthermore, the study of the distillation feature dimension should be in a separate table to show the effect of this feature dimension to the training speed and rendering quality.

- Many hyperparameter values, such as the depth-pushing loss weight and the threshold $\delta$, are set without any discussion. The sensitivity of these parameters, especially how they influence ray distribution and performance, needs exploration. For example, excessive depth-pushing loss weight may result in sub-optimal ray distributions.

- Including a plot of ray distribution with and without the depth-pushing loss would better illustrate its effectiveness.

- The approach’s core relies on the view-consistency metric, which is affected significantly by the experimentally determined threshold $\delta$. Several questions arise:
  - Does the threshold vary significantly across different datasets?
  - How sensitive is $\delta$ to performance?
  - Does the threshold vary across different ray locations?

- The paper only evaluates the proposed approach on Nerfacto, making it unclear if this method could improve other models, though it appears applicable to any NeRF-based methods.

- Considering the method’s reliance on DINOv2 features, it would be insightful to examine whether features from other foundation models or weaker models could achieve similar performance. In other words, are DINOv2 features truly indispensable for this method?

**Questions:**

Please refer to the Weaknesses section. I combined Weaknesses and Questions since they are linked and need to be together for better clarity.

---

### Official Review · Reviewer_fzaD · 2024-11-03

**Soundness:** 3
**Presentation:** 3
**Contribution:** 2
**Rating:** 3
**Confidence:** 5

**Summary:**

This paper introduces two components to enhance the training of Neural Radiance Fields (NeRF): a point sampling method and a loss function. The approach involves projecting a 3D point onto multiple views and calculating feature similarity in the image space to assess the likelihood of the point lying on the object surface. This prior knowledge is leveraged for point sampling during the fine sampling stage. Additionally, a depth-pushing loss is implemented to prevent background collapse. The proposed methods are evaluated on two widely-used datasets, demonstrating improvements over NeRF-related baselines. However, the focus is on novel view rendering, but the paper does not mention or compare its methods to 3D Gaussian Splatting (3DGS). In my perspective, the paper is not ready for publishing.

**Strengths:**

1. The paper is well-organized and easy to follow.

2. Visualizations are clear and effective, aiding in understanding the concepts and improvements in visual quality.

3. Results are evaluated on two datasets with varying numbers of input images, consistently showing that the proposed method outperforms previous NeRF baselines.

4. The approach of determining surface points through feature similarity across multiple views is intuitive and promising.

**Weaknesses:**

1. In the original NeRF framework, a coarse MLP is used to estimate the density of sampled points for importance sampling. The proposed method replaces this with the feature similarity metric for weight computation, which, while effective, may not be as novel as claimed. It acts as an alternative rather than a completely new sampling method.

2. The proposed depth-pushing loss is conceptually similar to the distortion loss in MipNeRF360. A more detailed comparison and discussion of these approaches would be valuable.

3. Given the objective of novel view synthesis, it is surprising that the authors do not reference or compare their work with 3D Gaussian Splatting, which is significantly faster and achieves better performance than NeRF-based methods. While this does not imply that NeRF research is obsolete, a fair comparison and discussion would provide context and strengthen the research by situating it within the broader field of computer vision.

4. Although the proposed evaluation settings are useful for assessing performance with different input counts, they do not facilitate comparisons with many previous baselines. The authors should report results using the standard dataset with all available images and benchmark their method against more existing approaches, including 3DGS-based methods.

**Questions:**

See weakness

---

### Official Review · Reviewer_CGmW · 2024-11-03

**Soundness:** 4
**Presentation:** 4
**Contribution:** 3
**Rating:** 6
**Confidence:** 4

**Summary:**

This paper trains a discriminative feature extractor based on an image foundation model and measures the multi-view consistency of sampling points by constructing a metric similar to the cost volume. The sampling process is then guided to focus more on points with better multi-view consistency through importance sampling. With this proposed method, the generated NeRF representations are constrained to have more accurate geometry.

**Strengths:**

- The proposed approach is both interesting and methodologically sound.

- The paper is well-written, with a clear and well-motivated idea.

- Experimental results demonstrate superior performance compared to existing methods.

**Weaknesses:**

- According to Table 1, the proposed method appears to significantly increase the training time.

- There is a typographical error: duplicated "the" in Line 270.

**Questions:**

- Is the proposed sampling method needed only during training, or during both training and inference?

- Figure 5 only reports performance when training views are more than 10. I wonder if the proposed method still works with fewer training views, such as 3 and 5.

---

### Official Review · Reviewer_Zszv · 2024-11-04

**Soundness:** 3
**Presentation:** 3
**Contribution:** 2
**Rating:** 3
**Confidence:** 5

**Summary:**

This paper proposes a sampling method for NeRF training. The authors use high-level distilled features and low-level RGB values, to compute the view-consistency metrics for points along the ray. Based on the metrics, they sample the points for NeRF training. A straightforward depth-pushing loss is further proposed to favor distant samples and prevent background collapse. The paper is easy to understand.

**Strengths:**

The paper is well-written and easy to understand. The motivation is reasonable and straightforward. The authors conduct comprehensive experiments on NeRF to demonstrate its effectiveness.

**Weaknesses:**

1. The proposed method cannot be applied to 3D Gaussian splatting, which is much faster in training and rendering compared with NeRF. The authors should discuss how their approach might be adapted to 3DGS or explain why they still chose to focus on NeRF.

2. Using high-level features and low-level RGBs to calculate the view-consistency metrics is not novel. Many former works on learning-based feature matching have explored this before. For example, [1] SuperGlue: Learning Feature Matching with Graph Neural Networks, [2] GIM: Learning Generalizable Image Matcher From Internet Videos. The authors should clarify how the proposed method differs from existing approaches.

**Questions:**

The authors propose a point sampling method for NeRF. However, 3DGS has become the prevailing representation of the neural field. The authors should discuss how to adapt their method to 3DGS. Besides, using high-level features and low-level RGBs to calculate the view consistency metrics has been extensively explored in existing learning-based feature matching methods. The proposed method does not articulate clear contributions. Therefore, I think the novelty cannot meet the bar of ICLR.

---

### Note · Authors · 2024-11-14

I have read and agree with the venue's withdrawal policy on behalf of myself and my co-authors.